# Boltzmann Exploration Done Right

**Nicolò Cesa-Bianchi**
Università degli Studi di Milano
Milan, Italy
nicolo.cesa-bianchi@unimi.it

**Claudio Gentile**
INRIA Lille – Nord Europe
Villeneuve d'Ascq, France
cla.gentile@gmail.com

**Gábor Lugosi**
ICREA & Universitat Pompeu Fabra
Barcelona, Spain
gabor.lugosi@gmail.com

**Gergely Neu**
Universitat Pompeu Fabra
Barcelona, Spain
gergely.neu@gmail.com

## Abstract

Boltzmann exploration is a classic strategy for sequential decision-making under uncertainty, and is one of the most standard tools in Reinforcement Learning (RL). Despite its widespread use, there is virtually no theoretical understanding about the limitations or the actual benefits of this exploration scheme. Does it drive exploration in a meaningful way? Is it prone to misidentifying the optimal actions or spending too much time exploring the suboptimal ones? What is the right tuning for the learning rate? In this paper, we address several of these questions for the classic setup of stochastic multi-armed bandits. One of our main results is showing that the Boltzmann exploration strategy with any monotone learning-rate sequence will induce suboptimal behavior. As a remedy, we offer a simple non-monotone schedule that guarantees near-optimal performance, albeit only when given prior access to key problem parameters that are typically not available in practical situations (like the time horizon $T$ and the suboptimality gap $\Delta$). More importantly, we propose a novel variant that uses different learning rates for different arms, and achieves a distribution-dependent regret bound of order $\frac{K \log^2 T}{\Delta}$ and a distribution-independent bound of order $\sqrt{KT} \log K$ without requiring such prior knowledge. To demonstrate the flexibility of our technique, we also propose a variant that guarantees the same performance bounds even if the rewards are heavy-tailed.

## 1 Introduction

Exponential weighting strategies are fundamental tools in a variety of areas, including Machine Learning, Optimization, Theoretical Computer Science, and Decision Theory [3]. Within Reinforcement Learning [23, 25], exponential weighting schemes are broadly used for balancing exploration and exploitation, and are equivalently referred to as Boltzmann, Gibbs, or softmax exploration policies [22, 14, 24, 19]. In the most common version of Boltzmann exploration, the probability of choosing an arm is proportional to an exponential function of the empirical mean of the reward of that arm. Despite the popularity of this policy, very little is known about its theoretical performance, even in the simplest reinforcement learning setting of *stochastic bandit problems*.

The variant of Boltzmann exploration we focus on in this paper is defined by

$$p_{t,i} \propto e^{\eta_t \widehat{\mu}_{t,i}}, \tag{1}$$

where $p_{t,i}$ is the probability of choosing arm $i$ in round $t$, $\widehat{\mu}_{t,i}$ is the empirical average of the rewards obtained from arm $i$ up until round $t$, and $\eta_t > 0$ is the *learning rate*. This variant is broadly used

in reinforcement learning [23, 25, 14, 26, 16, 18]. In the multiarmed bandit literature, exponential-weights algorithms are also widespread, but they typically use *importance-weighted* estimators for the rewards —see, e.g., [6, 8] (for the nonstochastic setting), [12] (for the stochastic setting), and [20] (for both stochastic and nonstochastic regimes). The theoretical behavior of these algorithms is generally well understood. For example, in the stochastic bandit setting Seldin and Slivkins [20] show a regret bound of order $\frac{K \log^2 T}{\Delta}$, where $\Delta$ is the suboptimality gap (i.e., the smallest difference between the mean reward of the optimal arm and the mean reward of any other arm).

In this paper, we aim to achieve a better theoretical understanding of the *basic* variant of the Boltzmann exploration policy that relies on the empirical mean rewards. We first show that any monotone learning-rate schedule will inevitably force the policy to either spend too much time drawing suboptimal arms or completely fail to identify the optimal arm. Then, we show that a specific non-monotone schedule of the learning rates can lead to regret bound of order $\frac{K \log T}{\Delta^2}$. However, the learning schedule has to rely on full knowledge of the gap $\Delta$ and the number of rounds $T$. Moreover, our negative result helps us to identify a crucial shortcoming of the Boltzmann exploration policy: it does not reason about the uncertainty of the empirical reward estimates. To alleviate this issue, we propose a variant that takes this uncertainty into account by using separate learning rates for each arm, where the learning rates account for the uncertainty of each reward estimate. We show that the resulting algorithm guarantees a distribution-dependent regret bound of order $\frac{K \log^2 T}{\Delta}$, and a distribution-independent bound of order $\sqrt{KT} \log K$.

Our algorithm and analysis is based on the so-called *Gumbel–softmax* trick that connects the exponential-weights distribution with the maximum of independent random variables from the Gumbel distribution.

## 2 The stochastic multi-armed bandit problem

Consider the setting of stochastic multi-armed bandits: each arm $i \in [K] \stackrel{\text{def}}{=} \{1, 2, \ldots, K\}$ yields a reward with distribution $\nu_i$, mean $\mu_i$, with the optimal mean reward being $\mu^* = \max_i \mu_i$. Without loss of generality, we will assume that the optimal arm is unique and has index 1. The gap of arm $i$ is defined as $\Delta_i = \mu^* - \mu_i$. We consider a repeated game between the learner and the environment, where in each round $t = 1, 2, \ldots$, the following steps are repeated:

1. The learner chooses an arm $I_t \in [K]$,

2. the environment draws a reward $X_{t,I_t} \sim \nu_{I_t}$ independently of the past,

3. the learner receives and observes the reward $X_{t,I_t}$.

The performance of the learner is measured in terms of the *pseudo-regret* defined as

$$R_T = \mu^* T - \sum_{t=1}^{T} \mathbb{E}[X_{t,I_t}] = \mu^* T - \mathbb{E}\left[\sum_{t=1}^{T} \mu_{I_t}\right] = \mathbb{E}\left[\sum_{t=1}^{T} \Delta_{I_t}\right] = \sum_{i=1}^{K} \Delta_i \mathbb{E}[N_{T,i}], \quad (2)$$

where we defined $N_{t,i} = \sum_{s=1}^{t} \mathbb{I}_{\{I_s=i\}}$, that is, the number of times that arm $i$ has been chosen until the end of round $t$. We aim at constructing algorithms that guarantee that the regret grows sublinearly.

We will consider the above problem under various assumptions of the distribution of the rewards. For most of our results, we will assume that each $\nu_i$ is $\sigma$-*subgaussian* with a known parameter $\sigma > 0$, that is, that

$$\mathbb{E}\left[e^{y(X_{1,i} - \mathbb{E}[X_{1,i}])}\right] \leq e^{\sigma^2 y^2 / 2}$$

holds for all $y \in \mathbb{R}$ and $i \in [K]$. It is easy to see that any random variable bounded in an interval of length $B$ is $B^2/4$-subgaussian. Under this assumption, it is well known that any algorithm will suffer a regret of at least $\Omega\left(\sum_{i>1} \frac{\sigma^2 \log T}{\Delta_i}\right)$, as shown in the classic paper of Lai and Robbins [17]. There exist several algorithms guaranteeing matching upper bounds, even for finite horizons [7, 10, 15]. We refer to the survey of Bubeck and Cesa-Bianchi [9] for an exhaustive treatment of the topic.

# 3 Boltzmann exploration done wrong

We now formally describe the heuristic form of Boltzmann exploration that is commonly used in the reinforcement learning literature [23, 25, 14]. This strategy works by maintaining the empirical estimates of each $\mu_i$ defined as

$$\widehat{\mu}_{t,i} = \frac{\sum_{s=1}^{t} X_{s,i} \mathbb{I}_{\{I_s=i\}}}{N_{t,i}} \tag{3}$$

and computing the exponential-weights distribution (1) for an appropriately tuned sequence of *learning rate* parameters $\eta_t > 0$ (which are often referred to as the *inverse temperature*). As noted on several occasions in the literature, finding the right schedule for $\eta_t$ can be very difficult in practice [14, 26]. Below, we quantify this difficulty by showing that natural learning-rate schedules may fail to achieve near-optimal regret guarantees. More precisely, they may draw suboptimal arms too much even after having estimated all the means correctly, or commit too early to a suboptimal arm and never recover afterwards. We partially circumvent this issue by proposing an admittedly artificial learning-rate schedule that actually guarantees near-optimal performance. However, a serious limitation of this schedule is that it relies on prior knowledge of problem parameters $\Delta$ and $T$ that are typically unknown at the beginning of the learning procedure. These observations lead us to the conclusion that the Boltzmann exploration policy as described by Equations (1) and (3) is no more effective for regret minimization than the simplest alternative of $\varepsilon$-greedy exploration [23, 7].

Before we present our own technical results, we mention that Singh et al. [21] propose a learning-rate schedule $\eta_t$ for Boltzmann exploration that simultaneously guarantees that all arms will be drawn infinitely often as $T$ goes to infinity, and that the policy becomes greedy in the limit. This property is proven by choosing a learning-rate schedule adaptively to ensure that in each round $t$, each arm gets drawn with probability at least $\frac{1}{t}$, making it similar in spirit to $\varepsilon$-greedy exploration. While this strategy clearly leads to sublinear regret, it is easy to construct examples on which it suffers a regret of at least $\Omega\left(T^{1-\alpha}\right)$ for any small $\alpha > 0$. In this paper, we pursue a more ambitious goal: we aim to find out whether Boltzmann exploration can actually guarantee polylogarithmic regret. In the rest of this section, we present both negative and positive results concerning the standard variant of Boltzmann exploration, and then move on to providing an efficient generalization that achieves consistency in a more universal sense.

## 3.1 Boltzmann exploration with monotone learning rates is suboptimal

In this section, we study the most natural variant of Boltzmann exploration that uses a monotone learning-rate schedule. It is easy to see that in order to achieve sublinear regret, the learning rate $\eta_t$ needs to *increase* with $t$ so that the suboptimal arms are drawn with less and less probability as time progresses. For the sake of clarity, we study the simplest possible setting with two arms with a gap of $\Delta$ between their means. We first show that, in order to guarantee near-optimal (logarithmic) regret, the learning rate has to increase at least at a rate $\frac{\log t}{\Delta}$ even when the mean rewards are perfectly known, and that any learning-rate sequence that increases at a slower logarithmic rate will lead to polynomial regret. In other words, $\frac{\log t}{\Delta}$ is the minimal affordable learning rate.

**Proposition 1.** *Let us assume that $\widehat{\mu}_{t,i} = \mu_i$ for all $t$ and $i = 1, 2$ with $\mu_1 > \mu_2$. Assume that for some constants $k \geq 1$, $\alpha \geq 0$ and $\varepsilon \leq \frac{1}{\Delta}$, the learning rate satisfies $\eta_t \leq \frac{\log(t\Delta^2)}{(1+\alpha)\Delta} + \varepsilon$ for all $t \geq k$. Then, the regret grows as*

- $R_T = \Omega\left(\frac{\log T}{\Delta}\right)$ *if $\alpha = 0$, and*

- $R_T = \Omega\left(T^{\frac{\alpha}{1+\alpha}}\left(\frac{1}{\Delta}\right)^{\frac{1-\alpha}{1+\alpha}}\right)$ *if $\alpha > 0$.*

*Proof.* For $t \geq k$, the probability of pulling the suboptimal arm can be bounded as

$$\mathbb{P}\left[I_t = 2\right] = \frac{1}{1 + e^{\eta_t \Delta}} \geq \frac{e^{-\eta_t \Delta}}{2} = \Omega\left(\left(\Delta^2 t\right)^{-\frac{1}{1+\alpha}}\right)$$

by our assumption on $\eta_t$. Summing up for all $t$, we get that the regret is at least

$$R_T = \Delta \sum_{t=1}^{T} \mathbb{P}\left[I_t = 2\right] \geq \Delta \cdot \left(k + \Omega\left(\sum_{t=k}^{T}\left(\Delta^2 t\right)^{-\frac{1}{1+\alpha}}\right)\right).$$

The proof is concluded by observing that the sum $\sum_{t=k}^{T} t^{-\frac{1}{1+\alpha}}$ is of the order $\Omega\left(\log T\right)$ if $\alpha = 0$ and $\Omega\left(T^{\frac{\alpha}{1+\alpha}}\right)$ if $\alpha > 0$. $\qquad\square$

This simple proposition thus implies an asymptotic lower bound on the schedule of learning rates $\eta_t$ that provide near-optimal guarantees. In contrast, Theorem 1 below shows that all learning rate sequences that grow faster than $2\log t$ yield a linear regret, provided this schedule is adopted since the beginning of the game. This should be contrasted with Theorem 2, which exhibits a schedule achieving logarithmic regret where $\eta_t$ grows faster than $2\log t$ only after the first $\tau$ rounds.

**Theorem 1.** *There exists a 2-armed stochastic bandit problem with rewards bounded in $[0,1]$ where Boltzmann exploration using any learning rate sequence $\eta_t$ such that $\eta_t > 2\log t$ for all $t \geq 1$ has regret $R_T = \Omega(T)$.*

*Proof.* Consider the case where arm 2 gives a reward deterministically equal to $\frac{1}{2}$ whereas the optimal arm 1 has a Bernoulli distribution of parameter $p = \frac{1}{2} + \Delta$ for some $0 < \Delta < \frac{1}{2}$. Note that the regret of any algorithm satisfies $R_T \geq \Delta(T - t_0)\mathbb{P}\left[\forall t > t_0, \ I_t = 2\right]$. Without loss of generality, assume that $\widehat{\mu}_{1,1} = 0$ and $\widehat{\mu}_{1,2} = 1/2$. Then for all $t$, independent of the algorithm, $\widehat{\mu}_{t,2} = 1/2$ and

$$p_{t,1} = \frac{e^{\eta_t \mathrm{Bin}(N_{t-1,1},p)}}{e^{\eta_t/2} + e^{\eta_t \mathrm{Bin}(N_{t-1,1},p)}} \quad \text{and} \quad p_{t,2} = \frac{e^{\eta_t/2}}{e^{\eta_t/2} + e^{\eta_t \mathrm{Bin}(N_{t-1,1},p)}}.$$

For $t_0 \geq 1$, Let $E_{t_0}$ be the event that $\mathrm{Bin}(N_{t_0,1}, p) = 0$, that is, up to time $t_0$, arm 1 gives only zero reward whenever it is sampled. Then

$$\mathbb{P}\left[\forall t > t_0 \ I_t = 2\right] \geq \mathbb{P}\left[E_{t_0}\right]\left(1 - \mathbb{P}\left[\exists t > t_0 \ I_t = 1 \mid E_{t_0}\right]\right)$$

$$\geq \left(\frac{1}{2} - \Delta\right)^{t_0}\left(1 - \mathbb{P}\left[\exists t > t_0 \ I_t = 1 \mid E_{t_0}\right]\right).$$

For $t > t_0$, let $A_{t,t_0}$ be the event that arm 1 is sampled at time $t$ but not at any of the times $t_0 + 1, t_0 + 2, \ldots, t - 1$. Then, for any $t_0 \geq 1$,

$$\mathbb{P}\left[\exists t > t_0 \ I_t = 1 \mid E_{t_0}\right] = \mathbb{P}\left[\exists t > t_0 \ A_{t,t_0} \mid E_{t_0}\right] \leq \sum_{t>t_0}\mathbb{P}\left[A_{t,t_0} \mid E_{t_0}\right]$$

$$= \sum_{t>t_0}\frac{1}{1 + e^{\eta_t/2}}\prod_{s=t_0+1}^{t-1}\left(1 - \frac{1}{1 + e^{\eta_s/2}}\right) \leq \sum_{t>t_0}e^{-\eta_t/2}.$$

Therefore

$$R_T \geq \Delta(T - t_0)\left(\frac{1}{2} - \Delta\right)^{t_0}\left(1 - \sum_{t>t_0}e^{-\eta_t/2}\right).$$

Assume $\eta_t \geq c\log t$ for some $c > 2$ and for all $t \geq t_0$. Then

$$\sum_{t>t_0}e^{-\eta_t/2} \leq \sum_{t>t_0}t^{-\frac{c}{2}} \leq \int_{t_0}^{\infty}x^{-\frac{c}{2}}\,dx = \left(\frac{c}{2} - 1\right)t_0^{-\left(\frac{c}{2}-1\right)} \leq \frac{1}{2}$$

whenever $t_0 \geq (2a)^{\frac{1}{a}}$ where $a = \frac{c}{2} - 1$. This implies $R_T = \Omega(T)$. $\qquad\square$

### 3.2 A learning-rate schedule with near-optimal guarantees

The above negative result is indeed heavily relying on the assumption that $\eta_t > 2\log t$ holds since the beginning. If we instead start off from a constant learning rate which we keep for a logarithmic number of rounds, then a logarithmic regret bound can be shown. Arguably, this results in a rather simplistic exploration scheme, which can be essentially seen as an *explore-then-commit* strategy (e.g., [13]). Despite its simplicity, this strategy can be shown to achieve near-optimal performance if the parameters are tuned as a function the suboptimality gap $\Delta$ (although its regret scales at the suboptimal rate of $1/\Delta^2$ with this parameter). The following theorem (proved in Appendix A.1) states this performance guarantee.

**Theorem 2.** *Assume the rewards of each arm are in $[0, 1]$ and let $\tau = \frac{16eK \log T}{\Delta^2}$. Then the regret of Boltzmann exploration with learning rate $\eta_t = \mathbb{I}_{\{t < \tau\}} + \frac{\log(t\Delta^2)}{\Delta}\mathbb{I}_{\{t \geq \tau\}}$ satisfies*

$$R_T \leq \frac{16eK \log T}{\Delta^2} + \frac{9K}{\Delta^2}.$$

## 4   Boltzmann exploration done right

We now turn to give a variant of Boltzmann exploration that achieves near-optimal guarantees without prior knowledge of either $\Delta$ or $T$. Our approach is based on the observation that the distribution $p_{t,i} \propto \exp{(\eta_t \widehat{\mu}_{t,i})}$ can be equivalently specified by the rule $I_t = \arg\max_j \{\eta_t \widehat{\mu}_{t,j} + Z_{t,j}\}$, where $Z_{t,j}$ is a standard Gumbel random variable[1] drawn independently for each arm $j$ (see, e.g., Abernethy et al. [1] and the references therein). As we saw in the previous section, this scheme fails to guarantee consistency in general, as it does not capture the uncertainty of the reward estimates. We now propose a variant that takes this uncertainty into account by choosing different scaling factors for each perturbation. In particular, we will use the simple choice $\beta_{t,i} = \sqrt{C^2/N_{t,i}}$ with some constant $C > 0$ that will be specified later. Our algorithm operates by independently drawing perturbations $Z_{t,i}$ from a standard Gumbel distribution for each arm $i$, then choosing action

$$I_{t+1} = \arg\max_i \{\widehat{\mu}_{t,i} + \beta_{t,i} Z_{t,i}\}. \tag{4}$$

We refer to this algorithm as *Boltzmann–Gumbel exploration*, or, in short, BGE. Unfortunately, the probabilities $p_{t,i}$ no longer have a simple closed form, nevertheless the algorithm is very straightforward to implement. Our main positive result is showing the following performance guarantee about the algorithm.[2]

**Theorem 3.** *Assume that the rewards of each arm are $\sigma^2$-subgaussian and let $c > 0$ be arbitrary. Then, the regret of Boltzmann–Gumbel exploration satisfies*

$$R_T \leq \sum_{i=2}^{K} \frac{9C^2 \log_+^2 (T\Delta_i/c^2)}{\Delta_i} + \sum_{i=2}^{K} \frac{c^2 e^\gamma + 18C^2 e^{\sigma^2/2C^2} (1 + e^{-\gamma})}{\Delta_i} + \sum_{i=2}^{K} \Delta_i.$$

*In particular, choosing $C = \sigma$ and $c = \sigma$ guarantees a regret bound of*

$$R_T = O\left(\sum_{i=2}^{K} \frac{\sigma^2 \log^2(T\Delta_i^2/\sigma^2)}{\Delta_i}\right).$$

Notice that, unlike any other algorithm that we are aware of, Boltzmann–Gumbel exploration still continues to guarantee meaningful regret bounds even if the subgaussianity constant $\sigma$ is underestimated—although such misspecification is penalized exponentially in the true $\sigma^2$. A downside of our bound is that it shows a suboptimal dependence on the number of rounds $T$: it grows asymptotically as $\sum_{i>1} \log^2(T\Delta_i^2)/\Delta_i$, in contrast to the standard regret bounds for the UCB algorithm of Auer et al. [7] that grow as $\sum_{i>1} (\log T)/\Delta_i$. However, our guarantee improves on the distribution-independent regret bounds of UCB that are of order $\sqrt{KT \log T}$. This is shown in the following corollary.

**Corollary 1.** *Assume that the rewards of each arm are $\sigma^2$-subgaussian. Then, the regret of Boltzmann–Gumbel exploration with $C = \sigma$ satisfies $R_T \leq 200\sigma\sqrt{KT} \log K$.*

Notably, this bound shows optimal dependence on the number of rounds $T$, but is suboptimal in terms of the number of arms. To complement this upper bound, we also show that these bounds are tight in the sense that the $\log K$ factor cannot be removed.

**Theorem 4.** *For any $K$ and $T$ such that $\sqrt{K/T} \log K \leq 1$, there exists a bandit problem with rewards bounded in $[0, 1]$ where the regret of Boltzmann–Gumbel exploration with $C = 1$ is at least $R_T \geq \frac{1}{2}\sqrt{KT} \log K$.*

The proofs can be found in the Appendices A.5 and A.6. Note that more sophisticated policies are known to have better distribution-free bounds. The algorithm MOSS [4] achieves minimax-optimal $\sqrt{KT}$ distribution-free bounds, but distribution-dependent bounds of the form $(K/\Delta)\log(T\Delta^2)$ where $\Delta$ is the suboptimality gap. A variant of UCB using action elimination and due to Auer and Ortner [5] has regret $\sum_{i>1}\log(T\Delta_i^2)/\Delta_i$ corresponding to a $\sqrt{KT(\log K)}$ distribution-free bound. The same bounds are achieved by the Gaussian Thompson sampling algorithm of Agrawal and Goyal [2], given that the rewards are subgaussian.

We finally provide a simple variant of our algorithm that allows to handle heavy-tailed rewards, intended here as reward distributions that are not subgaussian. We propose to use technique due to Catoni [11] based on the *influence function*

$$\psi(x) = \begin{cases} \log\left(1 + x + x^2/2\right), & \text{for } x \geq 0, \\ -\log\left(1 - x + x^2/2\right), & \text{for } x \leq 0. \end{cases}$$

Using this function, we define our estimates as

$$\widehat{\mu}_{t,i} = \beta_{t,i} \sum_{s=1}^{t} \mathbb{I}_{\{I_s=i\}} \psi\left(\frac{X_{s,i}}{\beta_{t,i}N_{t,i}}\right)$$

We prove the following result regarding Boltzmann–Gumbel exploration run with the above estimates.

**Theorem 5.** *Assume that the second moment of the rewards of each arm are bounded uniformly as* $\mathbb{E}\left[X_i^2\right] \leq V$ *and let* $c > 0$ *be arbitrary. Then, the regret of Boltzmann–Gumbel exploration satisfies*

$$R_T \leq \sum_{i=2}^{K} \frac{9C^2 \log_+^2\left(T\Delta_i/c^2\right)}{\Delta_i} + \sum_{i=2}^{K} \frac{c^2 e^{\gamma} + 18C^2 e^{V/2C^2}\left(1 + e^{-\gamma}\right)}{\Delta_i} + \sum_{i=2}^{K} \Delta_i.$$

Notably, this bound coincides with that of Theorem 3, except that $\sigma^2$ is replaced by $V$. Thus, by following the proof of Corollary 1, we can show a distribution-independent regret bound of order $\sqrt{KT}\log K$.

## 5 Analysis

Let us now present the proofs of our main results concerning Boltzmann–Gumbel exploration, Theorems 3 and 5. Our analysis builds on several ideas from Agrawal and Goyal [2]. We first provide generic tools that are independent of the reward estimator and then move on to providing specifics for both estimators.

We start with introducing some notation. We define $\widetilde{\mu}_{t,i} = \widehat{\mu}_{t,i} + \beta_{t,i}Z_{t,i}$, so that the algorithm can be simply written as $I_t = \arg\max_i \widetilde{\mu}_{t,i}$. Let $\mathcal{F}_{t-1}$ be the sigma-algebra generated by the actions taken by the learner and the realized rewards up to the beginning of round $t$. Let us fix thresholds $x_i, y_i$ satisfying $\mu_i \leq x_i \leq y_i \leq \mu_1$ and define $q_{t,i} = \mathbb{P}\left[\widetilde{\mu}_{t,1} > y_i \mid \mathcal{F}_{t-1}\right]$. Furthermore, we define the events $E_{t,i}^{\widehat{\mu}} = \{\widehat{\mu}_{t,i} \leq x_i\}$ and $E_{t,i}^{\widetilde{\mu}} = \{\widetilde{\mu}_{t,i} \leq y_i\}$. With this notation at hand, we can decompose the number of draws of any suboptimal $i$ as follows:

$$\mathbb{E}\left[N_{T,i}\right] = \sum_{t=1}^{T} \mathbb{P}\left[I_t = i, E_{t,i}^{\widetilde{\mu}}, E_{t,i}^{\widehat{\mu}}\right] + \sum_{t=1}^{T} \mathbb{P}\left[I_t = i, \overline{E_{t,i}^{\widetilde{\mu}}}, E_{t,i}^{\widehat{\mu}}\right] + \sum_{t=1}^{T} \mathbb{P}\left[I_t = i, \overline{E_{t,i}^{\widehat{\mu}}}\right]. \quad (5)$$

It remains to choose the thresholds $x_i$ and $y_i$ in a meaningful way: we pick $x_i = \mu_i + \frac{\Delta_i}{3}$ and $y_i = \mu_1 - \frac{\Delta_i}{3}$. The rest of the proof is devoted to bounding each term in Eq. (5). Intuitively, the individual terms capture the following events:

- The first term counts the number of times that, even though the estimated mean reward of arm $i$ is well-concentrated and the additional perturbation $Z_{t,i}$ is not too large, arm $i$ was drawn instead of the optimal arm 1. This happens when the optimal arm is poorly estimated or when the perturbation $Z_{t,1}$ is not large enough. Intuitively, this term measures the interaction between the perturbations $Z_{t,1}$ and the random fluctuations of the reward estimate $\widehat{\mu}_{t,1}$ around its true mean, and will be small if the perturbations tend to be large enough and the tail of the reward estimates is light enough.

- The second term counts the number of times that the mean reward of arm $i$ is well-estimated, but it ends up being drawn due to a large perturbation. This term can be bounded independently of the properties of the mean estimator and is small when the tail of the perturbation distribution is not too heavy.
- The last term counts the number of times that the reward estimate of arm $i$ is poorly concentrated. This term is independent of the perturbations and only depends on the properties of the reward estimator.

As we will see, the first and the last terms can be bounded in terms of the *moment generating function* of the reward estimates, which makes subgaussian reward estimators particularly easy to treat. We begin by the most standard part of our analysis: bounding the third term on the right-hand-side of (5) in terms of the moment-generating function.

**Lemma 1.** *Let us fix any $i$ and define $\tau_k$ as the $k$'th time that arm $i$ was drawn. We have*

$$\sum_{t=1}^{T} \mathbb{P}\left[I_t = i, \overline{E_{t,i}^{\widehat{\mu}}}\right] \leq 1 + \sum_{k=1}^{T-1} \mathbb{E}\left[\exp\left(\frac{\widehat{\mu}_{\tau_k,i} - \mu_i}{\beta_{\tau_k,i}}\right)\right] \cdot e^{-\frac{\Delta_i \sqrt{k}}{3C}}.$$

Interestingly, our next key result shows that the first term can be bounded by a nearly identical expression:

**Lemma 2.** *Let us define $\tau_k$ as the $k$'th time that arm $1$ was drawn. For any $i$, we have*

$$\sum_{t=1}^{T} \mathbb{P}\left[I_t = i, E_{t,i}^{\widetilde{\mu}}, E_{t,i}^{\widehat{\mu}}\right] \leq \sum_{k=0}^{T-1} \mathbb{E}\left[\exp\left(\frac{\mu_1 - \widehat{\mu}_{\tau_k,1}}{\beta_{\tau_k,1}}\right)\right] e^{-\gamma - \frac{\Delta_i \sqrt{k}}{3C}}.$$

It remains to bound the second term in Equation (5), which we do in the following lemma:

**Lemma 3.** *For any $i \neq 1$ and any constant $c > 0$, we have*

$$\sum_{t=1}^{T} \mathbb{P}\left[I_t = i, \overline{E_{t,i}^{\widetilde{\mu}}}, E_{t,i}^{\widehat{\mu}}\right] \leq \frac{9C^2 \log_+^2\left(T\Delta_i^2/c^2\right) + c^2 e^{\gamma}}{\Delta_i^2}.$$

The proofs of these three lemmas are included in the supplementary material.

## 5.1 The proof of Theorem 3

For this section, we assume that the rewards are $\sigma$-subgaussian and that $\widehat{\mu}_{t,i}$ is the empirical-mean estimator. Building on the results of the previous section, observe that we are left with bounding the terms appearing in Lemmas 1 and 2. To this end, let us fix a $k$ and an $i$ and notice that by the subgaussianity assumption on the rewards, the empirical mean $\widetilde{\mu}_{\tau_k,i}$ is $\frac{\sigma}{\sqrt{k}}$-subgaussian (as $N_{\tau_k,i} = k$). In other words,

$$\mathbb{E}\left[e^{\alpha\left(\widehat{\mu}_{\tau_k,i} - \mu_i\right)}\right] \leq e^{\alpha^2 \sigma^2 / 2k}$$

holds for any $\alpha$. In particular, using this above formula for $\alpha = 1/\beta_{\tau_k,i} = \sqrt{\frac{k}{C^2}}$, we obtain

$$\mathbb{E}\left[\exp\left(\frac{\widehat{\mu}_{\tau_k,i} - \mu_i}{\beta_{\tau_k,i}}\right)\right] \leq e^{\sigma^2 / 2C^2}.$$

Thus, the sum appearing in Lemma 1 can be bounded as

$$\sum_{k=1}^{T-1} \mathbb{E}\left[\exp\left(\frac{\widehat{\mu}_{\tau_k,i} - \mu_i}{\beta_{\tau_k,i}}\right)\right] \cdot e^{-\frac{\Delta_i \sqrt{k}}{3C}} \leq e^{\sigma^2 / 2C^2} \sum_{k=1}^{T-1} e^{-\frac{\Delta_i \sqrt{k}}{3C}} \leq \frac{18C^2 e^{\sigma^2 / 2C^2}}{\Delta_i^2},$$

where the last step follows from the fact[3] that $\sum_{k=0}^{\infty} e^{c\sqrt{k}} \leq \frac{2}{c^2}$ holds for all $c > 0$. The statement of Theorem 3 now follows from applying the same argument to the bound of Lemma 2, using Lemma 3, and the standard expression for the regret in Equation (2). □

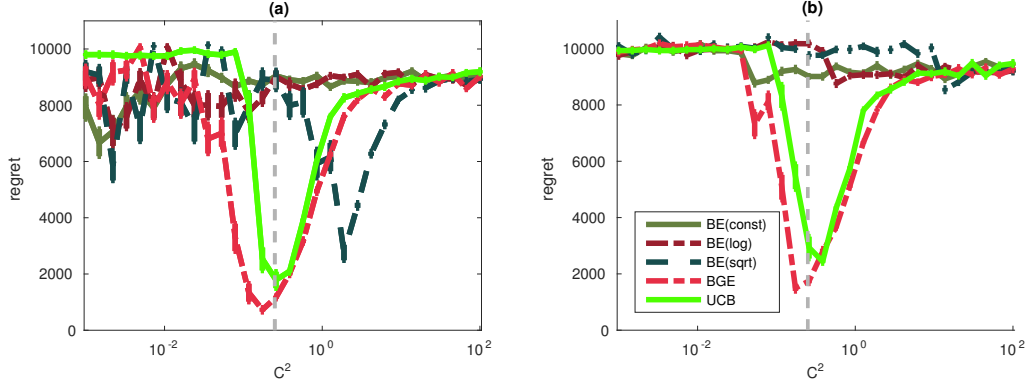

Figure 1: Empirical performance of Boltzmann exploration variants, Boltzmann–Gumbel exploration and UCB for (a) i.i.d. initialization and (b) malicious initialization, as a function of $C^2$. The dotted vertical line corresponds to the choice $C^2 = 1/4$ suggested by Theorem 3.

## 5.2 The proof of Theorem 5

We now drop the subgaussian assumption on the rewards and consider reward distributions that are possibly heavy-tailed, but have bounded variance. The proof of Theorem 5 trivially follows from the arguments in the previous subsection and using Proposition 2.1 of Catoni [11] (with $\theta = 0$) that guarantees the bound

$$\mathbb{E}\left[\exp\left(\pm\frac{\mu_i - \widehat{\mu}_{t,i}}{\beta_{t,i}}\right)\middle| N_{t,i} = n\right] \leq \exp\left(\frac{\mathbb{E}\left[X_i^2\right]}{2C^2}\right). \tag{6}$$

$\square$

## 6 Experiments

This section concludes by illustrating our theoretical results through some experiments, highlighting the limitations of Boltzmann exploration and contrasting it with the performance of Boltzmann–Gumbel exploration. We consider a stochastic multi-armed bandit problem with $K = 10$ arms each yielding Bernoulli rewards with mean $\mu_i = 1/2$ for all suboptimal arms $i > 1$ and $\mu_1 = 1/2 + \Delta$ for the optimal arm. We set the horizon to $T = 10^6$ and the gap parameter to $\Delta = 0.01$. We compare three variants of Boltzmann exploration with inverse learning rate parameters

- $\beta_t = C^2$ (BE-const),
- $\beta_t = C^2/\log t$ (BE-log), and
- $\beta_t = C^2/\sqrt{t}$ (BE-sqrt)

for all $t$, and compare it with Boltzmann–Gumbel exploration (BGE), and UCB with exploration bonus $\sqrt{C^2 \log(t)/N_{t,i}}$.

We study two different scenarios: (a) all rewards drawn i.i.d. from the Bernoulli distributions with the means given above and (b) the first $T_0 = 5,000$ rewards set to 0 for arm 1. The latter scenario simulates the situation described in the proof of Theorem 1, and in particular exposes the weakness of Boltzmann exploration with increasing learning rate parameters. The results shown on Figure 1 (a) and (b) show that while some variants of Boltzmann exploration may perform reasonably well when initial rewards take typical values and the parameters are chosen luckily, all standard versions fail to identify the optimal arm when the initial draws are not representative of the true mean (which happens with a small constant probability). On the other hand, UCB and Boltzmann–Gumbel exploration continue to perform well even under this unlikely event, as predicted by their respective theoretical guarantees. Notably, Boltzmann–Gumbel exploration performs comparably to UCB in this example (even slightly outperforming its competitor here), and performs notably well for the recommended parameter setting of $C^2 = \sigma^2 = 1/4$ (noting that Bernoulli random variables are $1/4$-subgaussian).

**Acknowledgements** Gábor Lugosi was supported by the Spanish Ministry of Economy and Competitiveness, Grant MTM2015-67304-P and FEDER, EU. Gergely Neu was supported by the UPFellows Fellowship (Marie Curie COFUND program n° 600387).

## Footnotes

[1]The cumulative density function of a standard Gumbel random variable is $F(x) = \exp(-e^{-x+\gamma})$ where $\gamma$ is the Euler-Mascheroni constant.

[2]We use the notation $\log_+(\cdot) = \max\{0, \cdot\}$.

[3]This can be easily seen by bounding the sum with an integral.

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
