[Supplementary Material · boltzmann_nips_finalfinal_full.pdf]

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

# A  Technical proofs

## A.1  The proof of Theorem 2

For any round $t$ and action $i$,

$$\frac{e^{-\eta_t}}{K} \le \frac{e^{\eta_t \widehat{\mu}_{t-1,i}}}{\sum_{j=1}^{K} e^{\eta_t \widehat{\mu}_{t-1,j}}} \le e^{\eta_t \left( \widehat{\mu}_{t-1,i} - \widehat{\mu}_{t-1,1} \right)} \ . \tag{7}$$

Now, for any $i > 1$, we can write

$$\mathbb{I}_{\{I_t=i\}} = \mathbb{I}_{\left\{ I_t=i, \, \widehat{\mu}_{t-1,i} - \widehat{\mu}_{t-1,1} < -\frac{\Delta_i}{2} \right\}} + \mathbb{I}_{\left\{ I_t=i, \, \widehat{\mu}_{t-1,i} - \widehat{\mu}_{t-1,1} \ge -\frac{\Delta_i}{2} \right\}}$$

$$\le \mathbb{I}_{\left\{ I_t=i, \, \widehat{\mu}_{t-1,i} - \widehat{\mu}_{t-1,1} < -\frac{\Delta_i}{2} \right\}} + \mathbb{I}_{\left\{ \widehat{\mu}_{t-1,1} \le \mu_1 - \frac{\Delta_i}{4} \right\}} + \mathbb{I}_{\left\{ \widehat{\mu}_{t-1,i} \ge \mu_i + \frac{\Delta_i}{4} \right\}} \ .$$

We take expectation of the three terms above and sum over $t = \tau + 1, \ldots, T$. Because of (7), the first term is simply bounded as

$$\sum_{t=\tau+1}^{T} \mathbb{P}\left[ I_t = i, \, \widehat{\mu}_{t-1,i} - \widehat{\mu}_{t-1,1} < -\frac{\Delta_i}{2} \right] \le \sum_{t=\tau+1}^{T} e^{-\eta_t \Delta_i / 2} \le \sum_{t=\tau+1}^{T} \frac{1}{t\Delta^2} \le \frac{\log(T+1)}{\Delta^2} \ .$$

We control the second and third term in the same way. For the second term we have that $\mathbb{I}_{\left\{ \widehat{\mu}_{t-1,1} \le \mu_1 - \frac{\Delta_i}{4} \right\}} \le \mathbb{I}_{\{N_{t-1,1} \le t_1\}} + \mathbb{I}_{\left\{ \widehat{\mu}_{t-1,1} \le \mu_1 - \frac{\Delta_i}{4}, \, N_{t-1,1} > t_1 \right\}}$ holds for any fixed $t$ and for any $t_1 \le t - 1$. Hence

$$\sum_{t=\tau+1}^{T} \mathbb{P}\left[ \widehat{\mu}_{t-1,1} \le \mu_1 - \frac{\Delta_i}{4} \right] \le \sum_{t=\tau+1}^{T} \mathbb{P}\left[ N_{t-1,1} \le t_1 \right] + \sum_{t=\tau+1}^{T} \mathbb{P}\left[ \widehat{\mu}_{t-1,1} \le \mu_1 - \frac{\Delta_i}{4}, \, N_{t-1,1} > t_1 \right] \ .$$

Now observe that, because of (7) applied to the initial $\tau$ rounds, $\mathbb{E}\left[ N_{t-1,1} \right] \ge \frac{\tau}{eK}$ holds for all $t > \tau$. By setting $t_1 = \frac{1}{2}\mathbb{E}\left[ N_{t-1,1} \right] \ge \frac{\tau}{2eK}$, Chernoff bounds (in multiplicative form) give $\mathbb{P}\left[ N_{t-1,1} \le t_1 \right] \le e^{-\frac{\tau}{8eK}}$. Standard Chernoff bounds, instead, give

$$\mathbb{P}\left[ \widehat{\mu}_{t-1,1} \le \mu_1 - \frac{\Delta_i}{4}, \, N_{t-1,1} > t_1 \right] \le \sum_{s=t_1+1}^{t-1} e^{-\frac{s\Delta^2}{8}} \le \frac{8}{\Delta^2} e^{-\frac{t_1 \Delta^2}{8}} \le \frac{8}{\Delta^2} e^{-\frac{\tau \Delta^2}{16eK}} \ .$$

Therefore, for the second term we can write

$$\sum_{t=\tau+1}^{T} \mathbb{P}\left[\widehat{\mu}_{t-1,1} \le \mu_1 - \frac{\Delta_i}{4}\right] \le T\left(e^{-\frac{\tau}{8eK}} + \frac{8}{\Delta^2}e^{-\frac{\tau\Delta^2}{16eK}}\right) \le 1 + \frac{8}{\Delta^2}\ .$$

The third term can be bounded exactly in the same way. Putting together, we have thus obtained, for all actions $i > 1$,

$$\sum_{i>1} \mathbb{E}\left[N_{T,i}\right] \le \tau + K + \frac{8K}{\Delta^2} \le \frac{16eK(\log T)}{\Delta^2} + \frac{9K}{\Delta^2}\ .$$

This concludes the proof. $\hfill\square$

## A.2 The proof of Lemma 1

Let $\tau_k$ denote the index of the round when arm $i$ is drawn for the $k$'th time. We let $\tau_0 = 0$ and $\tau_k = T$ for $k > N_{T,i}$. Then,

$$
\begin{aligned}
\sum_{t=1}^{T} \mathbb{P}\left[I_t = i, \overline{E_{t,i}^{\widehat{\mu}}}\right] &\le \mathbb{E}\left[\sum_{k=0}^{T-1} \sum_{t=\tau_k+1}^{\tau_{k+1}} \mathbb{I}_{\{I_t=i\}}\mathbb{I}_{\left\{\overline{E_{t,i}^{\widehat{\mu}}}\right\}}\right]\\
&= \mathbb{E}\left[\sum_{k=0}^{T-1} \mathbb{I}_{\left\{\overline{E_{\tau_k,i}^{\widehat{\mu}}}\right\}} \sum_{t=\tau_k+1}^{\tau_{k+1}} \mathbb{I}_{\{I_t=i\}}\right]\\
&= \mathbb{E}\left[\sum_{k=0}^{T-1} \mathbb{I}_{\left\{\overline{E_{\tau_k,i}^{\widehat{\mu}}}\right\}}\right]\\
&\le 1 + \sum_{k=1}^{T-1} \mathbb{P}\left[\widehat{\mu}_{\tau_k,i} \ge x_i\right]\\
&\le 1 + \sum_{k=1}^{T-1} \mathbb{P}\left[\widehat{\mu}_{\tau_k,i} - \mu_i \ge \frac{\Delta_i}{3}\right].
\end{aligned}
$$

Now, using the fact that $N_{\tau_k,i} = k$, we bound the last term by exploiting the subgaussianity of the rewards through Markov's inequality:

$$
\begin{aligned}
\mathbb{P}\left[\widehat{\mu}_{\tau_k,i} - \mu_i \ge \frac{\Delta_i}{3}\right] &= \mathbb{P}\left[e^{\alpha\left(\widehat{\mu}_{\tau_k,i}-\mu_i\right)} \ge e^{\alpha\frac{\Delta_i}{3}}\right] && \text{(for any } \alpha > 0)\\
&\le \mathbb{E}\left[e^{\alpha\left(\widehat{\mu}_{\tau_k,i}-\mu_i\right)}\right] \cdot e^{-\alpha\frac{\Delta_i}{3}} && \text{(Markov's inequality)}\\
&\le e^{\alpha^2\sigma^2/2k} \cdot e^{-\alpha\frac{\Delta_i}{3}} && \text{(the subgaussian property)}\\
&\le e^{\sigma^2/2C^2} \cdot e^{-\frac{\Delta_i\sqrt{k}}{3C}} && \text{(choosing } \alpha = \sqrt{k/C^2})
\end{aligned}
$$

Now, using the fact[4] that $\sum_{k=0}^{\infty} e^{c\sqrt{k}} \le \frac{2}{c^2}$ holds for all $c > 0$, the proof is concluded. $\hfill\square$

## A.3 The proof of Lemma 2

The proof of this lemma crucially builds on Lemma 1 of Agrawal and Goyal [2], which we state and prove below.

**Lemma 4** (cf. Lemma 1 of Agrawal and Goyal [2]).

$$\mathbb{P}\left[I_t = i, E_{t,i}^{\widehat{\mu}}, E_{t,i}^{\widetilde{\mu}}\,\Big|\, \mathcal{F}_{t-1}\right] \le \frac{1 - q_{t,i}}{q_{t,i}} \cdot \mathbb{P}\left[I_t = 1, E_{t,i}^{\widehat{\mu}}, E_{t,i}^{\widetilde{\mu}}\,\Big|\, \mathcal{F}_{t-1}\right]$$

*Proof.* First, note that $E_{t,i}^{\widehat{\mu}} \subseteq \mathcal{F}_{t-1}$. We only have to care about the case when $E_{t,i}^{\widetilde{\mu}}$ holds, otherwise both sides of the inequality are zero and the statement trivially holds. Thus, we only have to prove

$$\mathbb{P}\left[I_t = i \,\Big|\, \mathcal{F}_{t-1}, E_{t,i}^{\widetilde{\mu}}\right] \le \frac{1 - q_{t,i}}{q_{t,i}} \cdot \mathbb{P}\left[I_t = 1 \,\Big|\, \mathcal{F}_{t-1}, E_{t,i}^{\widetilde{\mu}}\right].$$

Now observe that $I_t = i$ under the event $E_{t,i}^{\widetilde{\mu}}$ implies $\widetilde{\mu}_{t,j} \le y_i$ for all $j$ (which follows from $\widetilde{\mu}_{t,j} \le \widetilde{\mu}_{t,i} \le y_i$). Thus, for any $i > 1$, we have

$$\begin{aligned}
\mathbb{P}\left[I_t = i \,\Big|\, \mathcal{F}_{t-1}, E_{t,i}^{\widetilde{\mu}}\right] &\le \mathbb{P}\left[\forall j : \widetilde{\mu}_{t,j} \le y_i \,\Big|\, \mathcal{F}_{t-1}, E_{t,i}^{\widetilde{\mu}}\right] \\
&= \mathbb{P}\left[\widetilde{\mu}_{t,1} \le y_i \,\Big|\, \mathcal{F}_{t-1}, E_{t,i}^{\widetilde{\mu}}\right] \cdot \mathbb{P}\left[\forall j > 1 : \widetilde{\mu}_{t,j} \le y_i \,\Big|\, \mathcal{F}_{t-1}, E_{t,i}^{\widetilde{\mu}}\right] \\
&= (1 - q_{t,i}) \cdot \mathbb{P}\left[\forall j > 1 : \widetilde{\mu}_{t,j} \le y_i \,\Big|\, \mathcal{F}_{t-1}, E_{t,i}^{\widetilde{\mu}}\right],
\end{aligned}$$

where the last equality holds because the event in question is independent of $E_{t,i}^{\widetilde{\mu}}$. Similarly,

$$\begin{aligned}
\mathbb{P}\left[I_t = 1 \,\Big|\, \mathcal{F}_{t-1}, E_{t,i}^{\widetilde{\mu}}\right] &\ge \mathbb{P}\left[\forall j > 1 : \widetilde{\mu}_{t,1} > y_i \ge \widetilde{\mu}_{t,j} \,\Big|\, \mathcal{F}_{t-1}, E_{t,i}^{\widetilde{\mu}}\right] \\
&= \mathbb{P}\left[\widetilde{\mu}_{t,1} > y_i \,\Big|\, \mathcal{F}_{t-1}, E_{t,i}^{\widetilde{\mu}}\right] \cdot \mathbb{P}\left[\forall j > 1 : \widetilde{\mu}_{t,j} \le y_i \,\Big|\, \mathcal{F}_{t-1}, E_{t,i}^{\widetilde{\mu}}\right] \\
&= q_{t,i} \cdot \mathbb{P}\left[\forall j > 1 : \widetilde{\mu}_{t,j} \le y_i \,\Big|\, \mathcal{F}_{t-1}, E_{t,i}^{\widetilde{\mu}}\right].
\end{aligned}$$

Combining the above two inequalities and multiplying both sides with $\mathbb{P}\left[E_{t,i}^{\widetilde{\mu}} \,\Big|\, \mathcal{F}_{t-1}\right]$ gives the result. $\qquad \square$

We are now ready to prove Lemma 2.

*Proof of Lemma 2.* Following straightforward calculations and using Lemma 4,

$$\sum_{t=1}^{T} \mathbb{P}\left[I_t = i, E_{t,i}^{\widetilde{\mu}}, E_{t,i}^{\widehat{\mu}}\right] \le \sum_{k=0}^{T-1} \mathbb{E}\left[\frac{1 - q_{\tau_k,i}}{q_{\tau_k,i}}\right].$$

Thus, it remains to bound the summands on the right-hand side. To achieve this, we start with rewriting $q_{\tau_k,i}$ as

$$\begin{aligned}
q_{\tau_k,i} = \mathbb{P}\left[\widetilde{\mu}_{\tau_k,1} > y_i \,|\, \mathcal{F}_{\tau_k-1}\right] &= \mathbb{P}\left[Z_{\tau_k,1} > \frac{\mu_1 - \widehat{\mu}_{\tau_k,1} - \frac{\Delta_i}{3}}{\beta_{\tau_k,1}} \,\Bigg|\, \mathcal{F}_{\tau_k-1}\right] \\
&= 1 - \exp\left(-\exp\left(-\frac{\mu_1 - \widehat{\mu}_{\tau_k,1} - \frac{\Delta_i}{3}}{\beta_{\tau_k,1}} + \gamma\right)\right),
\end{aligned}$$

so that we have

$$\begin{aligned}
\frac{1 - q_{\tau_k,i}}{q_{\tau_k,i}} &= \frac{\exp\left(-\exp\left(-\frac{\mu_1 - \widehat{\mu}_{\tau_k,1} - \frac{\Delta_i}{3}}{\beta_{\tau_k,1}} + \gamma\right)\right)}{1 - \exp\left(-\exp\left(-\frac{\mu_1 - \widehat{\mu}_{\tau_k,1} - \frac{\Delta_i}{3}}{\beta_{\tau_k,1}} + \gamma\right)\right)} \\
&\le \exp\left(\frac{\mu_1 - \widehat{\mu}_{\tau_k,1} - \frac{\Delta_i}{3}}{\beta_{\tau_k,1}} - \gamma\right) = \exp\left(\frac{\mu_1 - \widehat{\mu}_{t,1}}{\beta_{\tau_k,1}}\right) \cdot e^{-\gamma - \frac{\Delta_i}{3\beta_{\tau_k,1}}},
\end{aligned}$$

where we used the elementary inequality $\frac{e^{-1/x}}{1 - e^{-1/x}} \le x$ that holds for all $x \ge 0$. Taking expectations on both sides and using the definition of $\beta_{t,i}$ concludes the proof. $\qquad \square$

## A.4 Proof of Lemma 3

Setting $L = \frac{9C^2 \log^2\left(T\Delta_i^2/c^2\right)}{\Delta_i^2}$, we begin with the bound

$$\sum_{t=1}^{T} \mathbb{I}_{\left\{I_t = i, \overline{E_{t,i}^{\widetilde{\mu}}}, E_{t,i}^{\widehat{\mu}}\right\}} \leq L + \sum_{t=L}^{T} \mathbb{I}_{\left\{\widetilde{\mu}_{t,i} > \mu_1 - \frac{\Delta_i}{3}, \widehat{\mu}_{t,i} < \mu_i + \frac{\Delta_i}{3}, N_{t,i} > L\right\}}.$$

For bounding the expectation of the second term above, observe that

$$\mathbb{P}\left[\widetilde{\mu}_{t,i} > \mu_1 - \frac{\Delta_i}{3}, \widehat{\mu}_{t,i} < \mu_i + \frac{\Delta_i}{3}, N_{t,i} > L \,\middle|\, \mathcal{F}_{t-1}\right] \leq \mathbb{P}\left[\widetilde{\mu}_{t,i} > \widehat{\mu}_{t,i} + \frac{\Delta_i}{3}, N_{t,i} > L \,\middle|\, \mathcal{F}_{t-1}\right]$$

$$\leq \mathbb{P}\left[\beta_{t,i} Z_{t,i} > \frac{\Delta_i}{3}, N_{t,i} > L \,\middle|\, \mathcal{F}_{t-1}\right] = \mathbb{P}\left[Z_{t,i} > \frac{\Delta_i}{3\beta_{t,i}}, N_{t,i} > L \,\middle|\, \mathcal{F}_{t-1}\right].$$

By the distribution of the perturbations $Z_{t,i}$, we have

$$\mathbb{P}\left[Z_{t,i} \geq \frac{\Delta_i}{3\beta_{t,i}} \,\middle|\, \mathcal{F}_{t-1}\right] = 1 - \exp\left(-\exp\left(-\frac{\Delta_i}{3\beta_{t,i}} + \gamma\right)\right)$$

$$\leq \exp\left(-\frac{\Delta_i}{3\beta_{t,i}} + \gamma\right) = \exp\left(-\frac{\Delta_i\sqrt{N_{t,i}}}{3C} + \gamma\right),$$

where we used the inequality $1 - e^{-x} \leq x$ that holds for all $x$ and the definition of $\beta_{t,i}$. Noticing that $N_{t,i}$ is measurable in $\mathcal{F}_{t-1}$, we obtain the bound

$$\mathbb{P}\left[Z_{t,i} > \frac{\Delta_i}{3\beta_{t,i}}, N_{t,i} > L \,\middle|\, \mathcal{F}_{t-1}\right] \leq \exp\left(-\frac{\Delta_i\sqrt{N_{t,i}}}{3C} + \gamma\right) \cdot \mathbb{I}_{\{N_{t,i}>L\}},$$

$$\leq \exp\left(-\frac{\Delta_i\sqrt{L}}{3C} + \gamma\right) \cdot \mathbb{I}_{\{N_{t,i}>L\}} \leq \frac{c^2 e^\gamma}{T\Delta_i^2},$$

where the last step follows from using the definition of $L$ and bounding the indicator by 1. Summing up for all $t$ and taking expectations concludes the proof. $\square$

## A.5 The proof of Corollary 1

Following the arguments in Section 5.1, we can show that the number of suboptimal draws can be bounded as

$$\mathbb{E}\left[N_{T,i}\right] \leq 1 + \sigma^2 \frac{A + B\log^2(T\Delta_i^2/\sigma^2)}{\Delta_i^2}$$

for all arms $i$, with constants $A = e^\gamma + 18\sqrt{e}\left(1 + e^{-\gamma}\right)$ and $B = 9$. We can obtain a distribution-independent bound by setting a threshold $\Delta > 0$ and writing the regret as

$$R_T \leq \sigma^2 \sum_{i:\Delta_i > \Delta} \frac{A + B\log^2(T\Delta_i^2/\sigma^2)}{\Delta_i} + \Delta T$$

$$\leq \sigma^2 K \frac{A + B\log^2(T\Delta^2/\sigma^2)}{\Delta} + \Delta T \qquad \text{(since } \log^2(x^2)/x \text{ is monotone decreasing for } x \leq 1\text{)}$$

$$\leq \sigma\sqrt{TK} \frac{A + B\log^2(K\log^2 K)}{\log K} + \sigma\sqrt{TK}\log K \qquad \text{(setting } \Delta = \sigma\sqrt{K/T}\log K\text{)}$$

$$\leq \sigma\sqrt{TK} \frac{A + 2B\log^2(K)}{\log K} + \sigma\sqrt{TK}\log K \qquad \text{(using } 2\log\log(x) \leq \log(x)\text{)}$$

$$\leq \sigma\sqrt{TK}\log K\left(2B + A/\log K\right) + \sigma\sqrt{TK}\log K$$

$$\leq (2A + 2B + 1)\sigma\sqrt{TK}\log K,$$

where we used $\log K \geq \frac{1}{2}$ that holds for $K \geq 2$. The proof is concluded by noting that $2A + 2B + 1 \approx 187.63 < 200$. $\square$

## A.6 The proof of Theorem 4

The simple counterexample for the proof follows the construction of Section 3 of Agrawal and Goyal [2]. Consider a problem with deterministic rewards for each arm: the optimal arm 1 always gives a reward of $\Delta = \sqrt{\frac{K}{T}}C_1$ and all the other arms give rewards of 0. Define $B_{t-1}$ as the event that $\sum_{i=2}^{K} N_{t,i} \leq \frac{C_2\sqrt{KT}}{\Delta}$. Let us study two cases depending on the probability $\mathbb{P}[A_{t-1}]$: If $\mathbb{P}[A_{t-1}] \leq \frac{1}{2}$, we have

$$R_T \geq R_t \geq \mathbb{E}\left[\sum_i N_{t,i}\Delta \,\middle|\, \overline{A_{t-1}}\right] \cdot \frac{1}{2}. \geq \frac{1}{2}C_2\sqrt{KT}. \tag{8}$$

In what follows, we will study the other case when $\mathbb{P}[A_{t-1}] \geq \frac{1}{2}$. We will show that, under this assumption, a suboptimal arm will be drawn in round $t$ with at least constant probability. In particular, we have

$$
\begin{aligned}
\mathbb{P}[I_t \neq 1] &= \mathbb{P}[\exists i > 1 : \widetilde{\mu}_{t,1} < \widetilde{\mu}_{t,i}] \\
&\geq \mathbb{P}[\widetilde{\mu}_{t,1} < \mu_1, \exists i > 1 : \mu_1 < \widetilde{\mu}_{t,i}] \\
&\geq \mathbb{P}[\widetilde{\mu}_{t,1} < \mu_1, \exists i > 1 : \mu_1 < \widetilde{\mu}_{t,i}|\, A_{t-1}]\,\mathbb{P}[A_{t-1}] \\
&\geq \mathbb{E}\left[\mathbb{P}[\widetilde{\mu}_{t,1} < \mu_1, \exists i > 1 : \mu_1 < \widetilde{\mu}_{t,i}|\, \mathcal{F}_{t-1}, A_{t-1}]\right]\frac{1}{2} \\
&= \mathbb{E}\left[\mathbb{P}[\widetilde{\mu}_{t,1} < \mu_1|\, \mathcal{F}_{t-1}, A_{t-1}] \cdot \mathbb{P}[\exists i > 1 : \mu_1 < \widetilde{\mu}_{t,i}|\, \mathcal{F}_{t-1}, A_{t-1}]\right]\frac{1}{2} \\
&= \mathbb{E}\left[\mathbb{P}[Z_{t,1} < 0|\, \mathcal{F}_{t-1}, A_{t-1}] \cdot \mathbb{P}[\exists i > 1 : \Delta < \beta_{t,i}Z_{t,i}|\, \mathcal{F}_{t-1}, A_{t-1}]\right]\frac{1}{2}.
\end{aligned}
$$

To proceed, observe that $\mathbb{P}[Z_{t,1} < 0] \geq 0.1$ and

$$
\begin{aligned}
\mathbb{P}[\exists i > 1 : \Delta < \beta_{t,i}Z_{t,i}|\, \mathcal{F}_{t-1}, A_{t-1}] &= \mathbb{P}\left[\exists i > 1 : \Delta\sqrt{N_{t,i}} < Z_{t,i}\,\middle|\, \mathcal{F}_{t-1}, A_{t-1}\right] \\
&= 1 - \prod_{i>1}\exp\left(-\exp\left(-\Delta\sqrt{N_{t,i}} + \gamma\right)\right) \\
&= 1 - \exp\left(-\sum_{i>1}\exp\left(-\Delta\sqrt{N_{t,i}} + \gamma\right)\right) \\
&= 1 - \exp\left(-\sum_{i>1}\frac{K-1}{K-1}\exp\left(-\Delta\sqrt{N_{t,i}} + \gamma\right)\right) \\
&\geq 1 - \exp\left(-(K-1)\exp\left(-\Delta\sqrt{\sum_{i>1}\frac{N_{t,i}}{K-1}} + \gamma\right)\right) \quad \text{(by Jensen's inequality)} \\
&\geq 1 - \exp\left(-(K-1)\exp\left(-\Delta\sqrt{\frac{C_2\sqrt{KT}}{\Delta(K-1)}} + \gamma\right)\right) \\
&= 1 - \exp\left(-(K-1)\exp\left(-\Delta\sqrt{\frac{C_2 T}{C_1(K-1)}} + \gamma\right)\right) \\
&\geq 1 - \exp\left(-\exp\left(-C_1\sqrt{\frac{C_2}{C_1}} + \log(K-1) + \gamma\right)\right).
\end{aligned}
$$

Setting $C_2 = C_1 = \log K$, we obtain that whenever $\mathbb{P}[A_{t-1}] \geq \frac{1}{2}$, we have

$$
\begin{aligned}
\mathbb{P}[I_t \neq 1] &\geq 1 - \exp\left(-\exp\left(-\log K + \log(K-1) + \gamma\right)\right) \\
&\geq 1 - \exp\left(-\exp(\gamma)\right) \geq 0.83 > \frac{1}{2}.
\end{aligned}
$$

This implies that the regret of our algorithm is at least

$$\frac{1}{2}T\Delta = \frac{1}{2}\sqrt{TK}\log K.$$

Together with the bound of Equation (8) for the complementary case, this concludes the proof. □