[Reviews · NeurIPS 2017]

Reviewer 1



Pros: - A systematic study on the classical Boltzmann exploration heuristic in the context of multi-armed bandit. The results provide useful insights to the understanding of Boltzmann exploration and multi-armed bandits - The paper is clearly written Cons: - The technique is incremental, and the technical contribution to multi-armed bandit research is small. The paper studiee Boltzmann exploration heuristic for reinforcement learning, namely use empirical means and exponential weight to probabilistically select actions (arms) in the context of multi-armed bandit. The purpose of the paper is to achieve property theoretical understanding of the Boltzmann exploration heuristic. I view that the paper achieves this goal by several useful results. First, the authors show that the standard Boltzmann heuristic may not achieve good learning result, in fact, the regret could be linear, when using monotone learning rates. Second, the authors show that, if the learning rate remain constant for a logarithmic number of steps and then increase, the regret is close to the optimal one. This learning strategy is essentially explore-then-commit strategy, but the catch is that it needs to know the critical problem parameter \Delta and T, which are typically unknown. Third, the authors propose to generalize the Boltzmann exploration by allowing individual learning rates for different arms based on their certainty, and show that this leads to good regret bounds. The above serious of results provide good understanding of Boltzmann exploration. In particular, it provides the theoretical insight that the naive Boltzmann exploration lacks control on the uncertainty of arms, so may not preform well. This insight may be useful in the more general setting of reinforcement learning. The paper is in general well written and easy to follow. The technical novelty of the paper is incremental. The analysis are based on existing techniques. The new technical contribution to the multi-armed bandit research is likely to be small, since there are already a number of solutions achieving optimal or new optimal regret bounds. Minor comments: - line 243, log_+(\cdot) = min{0, \cdot}. Should it be max instead of min?

Reviewer 2



This paper revisits Boltzmann exploration in stochastic MAB. It discusses when Boltzmann exploration could be done wrong (Section 3), and also discusses how to do it right (Section 4). This paper is interesting and well-written in general. Boltzmann exploration is a commonly used exploration scheme in MAB and RL, and this paper might improve the RL community's understanding of this important exploration scheme. That said, I recommend to accept this paper. However, I only give a "marginal acceptance" for the following reasons: 1) One key insight of this paper is not new. It is well-known in the RL community that a naive implementation of the Boltzmann exploration could lead to bad performance, and the main reason is that Boltzmann exploration does not consider the uncertainty of the empirical reward estimates. This reduces the novelty of this paper. 2) This paper has proposed one way to fix this issue (Section 4, "Boltzmann exploration done right"). However, there are many other ways to fix this issue by using a different exploration schemes (e.g. UCB1, Thompson sampling). It is not clear to me why we should use the modified Boltzmann exploration scheme proposed in this paper. In particular, no experimental comparisons have been provided in this paper. One question: in the first equation of Theorem 3, the regret is O(log^2(T)/\Delta^2). However, in the second equation of that theorem, the regret becomes O(log^2(T)/\Delta). Please explain.

Reviewer 3



[I have read the other reviews and the author feedback, and I maintain my accept rating. Looking forward to seeing the additional expt comparisons the authors alluded to in their feedback.] This paper studies several variants of Boltzmann exploration for stochastic bandit problems theoretically and convincingly demonstrates when (and why) it is sub-optimal as well as simple modifications to the exploration strategy to achieve (poly)logarithmic regret. These modifications differ from the classic learning-rate schedules in reinforcement learning (e.g. Singh et al'00) that attenuate the temperature of the Boltzmann distribution adaptively. Instead, this paper introduces per-action "learning rates" (reminiscent of per-action confidence intervals in UCB strategies) to achieve low regret. It is debatable if the resulting (implicit) distribution over arms looks anything like a Boltzmann distribution. A small experiment (e.g. a simulation study) will be very informative -- does this strategy remain robust when we get the scale of \beta's wrong (by poorly setting C)? does attenuated Boltzmann exploration (e.g. Singh et al'00) lose substantially to this approach for practical problem sizes? Since this approach precludes the use of importance weighted estimators (computing the posterior prob. of picking an arm seems to be hard) whereas attenuated Boltzmann exploration can also be used with importance weighting, a more careful comparison seems necessary. Minor: 113: Use >= in the eqns Does the analysis of Eqn(6) also apply to importance-weighted reward estimates (thereby providing a unified treatment of Seldin&Slivkins'14 also)?